# Heat Transport Exploration for Hybrid Nanoparticle (Cu, Fe_3_O_4_)—Based Blood Flow via Tapered Complex Wavy Curved Channel with Slip Features

**DOI:** 10.3390/mi13091415

**Published:** 2022-08-28

**Authors:** A. Abbasi, W. Farooq, El Sayed Mohamed Tag-ElDin, Sami Ullah Khan, M. Ijaz Khan, Kamel Guedri, Samia Elattar, M. Waqas, Ahmed M. Galal

**Affiliations:** 1Department of Mathematics, University of Azad Jammu and Kashmir Muzaffarabad, Muzaffarabad 13100, Pakistan; 2Faculty of Engineering and Technology, Future University in Egypt, New Cairo 11835, Egypt; 3Department of Mathematics, COMSATS University Islamabad, Sahiwal 57000, Pakistan; 4Department of Mathematics and Statistics, Riphah International University I-14, Islamabad 44000, Pakistan; 5Department of Mechanical Engineering, Lebanese American University, Beirut 2100, Lebanon; 6Mechanical Engineering Department, College of Engineering and Islamic Architecture, Umm Al-Qura University, P.O. Box 5555, Makkah 21955, Saudi Arabia; 7Research Unity: Materials, Energy and Renewable Energies, Faculty of Science of Gafsa, University of Gafsa, Gafsa 2100, Tunisia; 8Department of Industrial & Systems Engineering, College of Engineering, Princess Nourah bint Abdulrahman University, P.O. Box 84428, Riyadh 11671, Saudi Arabia; 9NUTECH School of Applied Sciences and Humanities, National University of Technology, Islamabad 44000, Pakistan; 10Mechanical Engineering Department, College of Engineering, Prince Sattam Bin Abdulaziz University, Wadi Addawaser 11991, Saudi Arabia; 11Production Engineering and Mechanical Design Department, Faculty of Engineering, Mansoura University, Mansoura 35516, Egypt

**Keywords:** Casson hybrid nanoparticles, peristaltic transport, slip effects, hall applications, numerical approach

## Abstract

Curved veins and arteries make up the human cardiovascular system, and the peristalsis process underlies the blood flowing in these ducts. The blood flow in the presence of hybrid nanoparticles through a tapered complex wavy curved channel is numerically investigated. The behavior of the blood is characterized by the Casson fluid model while the physical properties of iron (Fe_3_O_4_) and copper (Cu) are used in the analysis. The fundamental laws of mass, momentum and energy give rise the system of nonlinear coupled partial differential equations which are normalized using the variables, and the resulting set of governing relations are simplified in view of a smaller Reynolds model approach. The numerical simulations are performed using the computational software Mathematica’s built-in ND scheme. It is noted that the velocity of the blood is abated by the nanoparticles’ concentration and assisted in the non-uniform channel core. Furthermore, the nanoparticles’ volume fraction and the dimensionless curvature of the channel reduce the temperature profile.

## 1. Introduction

In various thermal techniques, the improvement of heat transfer with the proper utilization of nanofluids has emerged as a superior mechanism. Despite the continuous work in nanotechnology and thermal engineering reporting different means of enhancing the thermal processes, the results of increasing thermal resources have been notably poor due to high casting and low thermal performance. The nanoparticles in question are small sized metallic materials with good thermal accuracy. Recently, nanofluids have been used in different applications such as energy production, various engineering phenomena, as industrial resources, for thermal management, etc. In recent years, nanoparticles have been widely used in anti-cancer drugs to kill tumor cells and due to their enhanced thermal conductivity these nanoparticles destruct the tumor tissue more efficiently. In cancer treatment, nanoparticles are inserted into the bloodstream to increase therapeutic efficacy and reduce side effects; better performance can be obtained by inserting different combinations of nanoparticles. Compelling improvements in the thermal properties of nanofluids are a fundamental source of motivation. Choi [1] performed the directions first on nanofluids; Buongiorno [2] endorsed the Brownian motion and the role of thermophoretic forces on nanofluid flows; Sui et al. [3] observed heat transfer fluctuation for the slip flow of nanofluids with a rate-type Maxwell model; Sandeep and Animasaun [4] conducted research to highlight the change in thermal properties of nanofluid when variable sources of thermal properties are followed; and Afridi et al. [5] reflected the heat transfer onset of nanofluids in view of fractional heat sources. The convective thermal case associated with the couple stress nanofluid has been analytically reported in the contribution of Khan et al. [6]. Kumar et al. [7] reported the entropy generation involvement for the Joule heating flow in the presence of radiative phenomenon. Khan et al. [8] preserved the microorganisms’ suspension in nanofluids and inspected the stability analysis. The variable viscosity utilization for observing the nanofluid properties was noted by Mondal and Pal [9]. Ahmad et al. [10] determined the bioconvection investigation in a porous medium with nanoparticles’ attention. Makinde et al. [11] predicted the thermal investigation based on nanofluids with magneto-hemodynamic applications. Das et al. [12] focused on the fully developed vertically moving channel flow of nanofluids under the influence of wall surface conductivity. Mandal et al. [13] worked out a nanofluid model for carbon nanotubes comprising the rotating 3-D flow. Zhang et al. [14] discussed the bioconvective attribution of nanofluids in concentric cylinders with a dominant Lorentz force contribution. The Walter-B nanofluid flow with attention to buoyancy forces has been analyzed in the work of Chu et al. [15].

The hybrid nanofluid is a superior category of nanofluids which reflects the improved thermal consequences of the suspension of two different nanoparticles in a base material. The motivation to observe the thermal mechanism of the base fluid via the hybrid nanofluid is due to the enhanced thermal mechanism. The reflection of hybrid nanofluid models exhibit more fascinating behavior than nanofluids. Special applications of hybrid nanofluids are observed in energy production, manufacturing systems, solar applications, heating devices, engine heat rate countering, extrusion processes, etc. The class of hybrid nanofluids is achieved after combining the base material with more than one nanoparticle. The thermal onset of hybrid particles is more stable and impressive. Wahid et al. [16] observed the thermal attention of hybrid nanofluids to a moving disk with permeable surface walls. Almaneea et al. [17] focused on the comparative enhancement of the thermal aspect of the heating phenomenon in view of hybrid nanofluid interaction. Mousavi et al. [18] investigated the flow of a stretching porous space encountering the hybrid nanofluid model and detected some dual numerical solutions in a confined region. The thermal effects of a hybrid nanofluid under the impact of a magnetic force were visualized by Khan et al. [19]. Madhukesh et al. [20] determined the curved geometry flow due to the hybrid nanofluid by observing the Newtonian heating effects. Rashid et al. [21] analyzed the heating aspect of titanium oxide in a moving cylinder in the horizontal direction. Abdelmalek et al. [22] identified the rotation of the disk flow where the improvement in heat transfer was influenced by hybrid nanoparticles. Muhammad et al. [23] predicted the attention of hybrid nanofluid to the squeezing flow. Shaw et al. [24] reported the contribution of the quadratic radiative phenomenon while working on Casson hybrid nanoparticles for various values of the Prandtl number. Sen et al. [25] developed tiny hybrid nanoparticles of thermal significance with a diamond base material. Nayak et al. [26] observed a distinct nanostructure for non-Newtonian fluids. The enrollment of interesting Darcy–Forchheimer forces for optimized nanofluid flow was addressed by Nayak et al. [27]. Recently, the work of Shaw [28] presented the various novel consequences of a hybrid nanofluid model for rotatory disk flow. Mahanthesh et al. [29] observed the importance of Joule heating for hybrid nanofluid transport in a wedge capturing the isothermal properties. The permeable surface flow of hybrid nanofluid with theoretical observations has been determined by Haq et al. [30].

The application of the Hall effect is associated with direct current and is important in plasma physics, magnetic devices, electromagnetic theory, semi-conductors, voltage currents, circuit problems and various electrically conducting flows. The Hall current attains direct association with the magnetic force. Seth et al. [31] addressed the Hall effect for rotating flow in a channel owing to arbitrary conducting walls. Seth et al. [32] observed the Hall effect for the heat transfer phenomenon in ramped thermal temperature. Abbasi et al. [33] addressed the Hall effect on Jeffrey nanofluid caused by peristaltic transport.

The motivation for presenting the current flow model is the observation of the thermal impact of a hybrid nanofluid model for the slip flow of Casson fluid with applications of peristaltic phenomena [34,35,36]. The motivation for the consideration of the Casson fluid model is justified by the fact that it reports the shear thinning and shear thickening effects associated with human blood. The novel aspects of the current research are:➢The presentation of a mathematical model for the peristaltic transport of Casson fluid with the interaction of hybrid nanofluid containing the ferro nanoparticles and copper nanomaterials in a curved channel.➢The role of slip effects and Hall current is also observed.➢The highly nonlinear system of the obtained model is numerically solved with the ND-Solver.➢The physical thermal impact of hybrid nanoparticles is focused to control the blood flow properties. The current investigation presents novel applications for human blood flow, thermal systems, various engineering processes, extrusion systems, human endoscopy, the control of heating phenomena, chemical processes and biomedical applications [37,38,39,40,41,42].

## 2. Mathematical Modeling of Hybrid Nanofluid

Consider an unsteady base in a channel with a curved surface caused by 2-D flow, where the channel width is taken as 2b. The rheology of hybrid nanofluid in the curved region is due to peristaltic pumping. The complex sinusoidal waves with speed c are imposed on the walls of the conduit. In order to model the problem in the curved channel, curvilinear coordinates are incorporated instead of cylindrical coordinates. A sketch of the flow under conditions of the flow problem and coordinate system is presented in Figure 1. The transportation of fluid is chosen along the S-axis and the X-axis is chosen to be normal to the surface of the conduit. The radius of the channel is Λ and O is the center of the channel. 

The mathematical expressions to describe the complex peristaltic waves are [34,35,36]:(1)$$\bar{H_{2}}\left(\bar{S},\bar{t}\right)=b+\bar{M}\left(\bar{S}-c\bar{t}\right)+{\bar{\Phi }}_{1}\sin\left(\frac{2\sigma \pi }{\beta }\left(\bar{S}-c\bar{t}\right)\right)+{\bar{\Phi }}_{2}\sin\left(\frac{2\omega \pi }{\beta }\left(\bar{S}-c\bar{t}\right)\right)$$
(2)$${\bar{H}}_{1}\left(\bar{S},\bar{t}\right)=-b-\bar{M}\left(\bar{S}-c\bar{t}\right)-{\bar{\Phi }}_{1}\sin\left(\frac{2\sigma \pi }{\beta }\left(\bar{S}-c\bar{t}\right)+\sigma \epsilon \right)-{\bar{\Phi }}_{2}\sin\left(\frac{2\omega \pi }{\beta }\left(\bar{S}-c\bar{t}\right)+\omega \epsilon \right),$$
with wavelength (β), non-uniform factor $\left(\bar{M}\right)$, wave speed (c), phase difference (ϵ) and wave amplitudes $(\Phi_{1}$, $\Phi_{2}).$ The tangential, radial and axial directions are denoted with $\bar{X},$ $\bar{Z}$ and $\bar{S}$, respectively. The observations for a straight channel are observed when the curvature of channel Λ approaches to ∞. The axial and radial velocity components are ${\bar{U}}_{2}$ and ${\bar{U}}_{1}$, respectively.

The contribution of magnetic force is taken in radial directions with Lorentz force:(3)$$\bar{F}=\bar{J_{1}}\times \bar{B}$$

The Hall current is:(4)$$\bar{J_{1}}+\frac{en_{e}}{B_{0}}\left(\bar{J_{1}}\times \bar{B,}\right)=\sigma_{hnf}\left[\bar{V}\times \bar{B}\right]$$
having $n_{e}$ (free electron density), e (electric charge) and $\sigma_{hnf}$ (hybrid nanoparticles’ electric conductivity). Writing Equation (4) with components:(5)$$\begin{array}{c} J_{1r}=0,\\ \frac{m\Lambda }{\bar{X}+\Lambda }J_{1z}+J_{1\theta }=0,\\ -\frac{m\Lambda }{\bar{X}+\Lambda }J_{1\theta }+J_{1z}=-\frac{\sigma_{hnf}B_{0}\Lambda }{\bar{X}+\Lambda }{\bar{U}}_{2}, \end{array}\}$$
where, $m=en_{e}$ is the Hall parameter. In view of Equation (5):(6)$$J_{1\theta }=-\frac{\sigma_{hnf}B_{0}\Lambda^{2}}{{\left(\bar{X}+\Lambda \right)}^{2}}{\bar{U}}_{2}\left(\frac{m}{1+{\left(\frac{m\Lambda }{\bar{X}+\Lambda }\right)}^{2}}\right)$$
(7)$$J_{1z}=\frac{\sigma_{hnf}B_{0}\Lambda }{\left(\;\bar{X}+\Lambda \right)}{\bar{U}}_{2}\left(\frac{1}{1+{\left(\frac{m\Lambda }{\bar{X}+\Lambda }\right)}^{2}}\right)$$

All defined flow assumptions lead to following governing system [34,35,36]:(8)$$\frac{\partial }{\partial \bar{X}}\left\{\left(\bar{X}+\Lambda \right){\bar{U}}_{1}\right\}+\Lambda \frac{\partial {\bar{U}}_{2}}{\partial \bar{S}}=0$$
(9)$$\rho_{hnf}\left(\frac{\partial {\bar{U}}_{1}}{\partial \bar{t}}+U_{1}\frac{\partial {\bar{U}}_{1}}{\partial \bar{X}}+\frac{\bar{U_{2}}\Lambda }{\bar{X}+\Lambda }\frac{\partial {\bar{U}}_{1}}{\partial \bar{S}}-\frac{{\bar{U}}_{2}^{2}}{\bar{X}+\Lambda }\right)=\;-\frac{\partial \bar{P}}{\partial \bar{X}}+\mu_{hnf}\left(1+\frac{1}{\gamma_{1}}\right)\left(\frac{1}{\bar{X}+\Lambda }\frac{\partial }{\partial \bar{X}}\left(\left(\bar{X}+\Lambda \right)\frac{\partial {\bar{U}}_{1}}{\partial \bar{X}}\right)+{\left(\frac{\Lambda }{\bar{X}+\Lambda }\right)}^{2}\frac{\partial^{2}{\bar{U}}_{1}}{\partial {\bar{S}}^{2}}-\frac{2\Lambda }{{\left(\bar{X}+\Lambda \right)}^{2}}\frac{\partial {\bar{U}}_{2}}{\partial \bar{S}}-\frac{{\bar{U}}_{1}}{{\left(\bar{X}+\Lambda \right)}^{2}}\right)$$
(10)$$\rho_{hnf}\left(\frac{\partial {\bar{U}}_{2}}{\partial \bar{t}}+{\bar{U}}_{1}\frac{\partial {\bar{U}}_{2}}{\partial \bar{X}}+\frac{{\bar{U}}_{2}\Lambda }{\bar{X}+\Lambda }\frac{\partial {\bar{U}}_{2}}{\partial \bar{S}}-\frac{{\bar{U}}_{1}{\bar{U}}_{2}}{\bar{X}+\Lambda }\right)=\;-\frac{\Lambda }{\bar{X}+\Lambda }\frac{\partial \bar{P}}{\partial \bar{S}}+\mu_{hnf}\left(1+\frac{1}{\gamma_{1}}\right)\left(\frac{1}{\bar{X}+\Lambda }\frac{\partial }{\partial \bar{X}}\left(\left(\bar{X}+\Lambda \right)\frac{\partial {\bar{U}}_{2}}{\partial \bar{X}}\right)+{\left(\frac{\Lambda }{\bar{X}+\Lambda }\right)}^{2}\frac{\partial^{2}{\bar{U}}_{2}}{\partial {\bar{S}}^{2}}+\frac{2\Lambda }{{\left(\bar{X}+\Lambda \right)}^{2}}\frac{\partial {\bar{U}}_{1}}{\partial \bar{S}}-\frac{{\bar{U}}_{2}}{{\left(\bar{X}+\Lambda \right)}^{2}}\right)\;-\frac{\sigma_{hnf}B_{0}^{2}\Lambda^{2}}{{\left(\bar{X}+\Lambda \right)}^{2}}{\bar{U}}_{2}\left(\frac{1}{1+{\left(\frac{m\Lambda }{\bar{X}+\Lambda }\right)}^{2}}\right)$$
(11)$${\left(\rho C_{p}\right)}_{hnf}\left(\frac{\partial \bar{T}}{\partial \bar{t}}+{\bar{U}}_{1}\frac{\partial \bar{T}}{\partial \bar{X}}+\frac{{\bar{U}}_{2}\Lambda }{\bar{X}+\Lambda }\frac{\partial \bar{T}}{\partial \bar{S}}\right)=\;K_{\mathrm{hnf}}\left(\frac{1}{\bar{X}+\Lambda }\frac{\partial }{\partial \bar{X}}\left(\left(\bar{X}+\Lambda \right)\frac{\partial \bar{T}}{\partial \bar{X}}\right)+{\left(\frac{\Lambda }{\bar{X}+\Lambda }\right)}^{2}\frac{\partial^{2}\bar{T}}{\partial {\bar{S}}^{2}}\right)\;+\mu_{hnf}\left(1+\frac{1}{\gamma_{1}}\right)\left(2\left\{{\left(\frac{\partial {\bar{U}}_{1}}{\partial \bar{X}}\right)}^{2}+{\left(\frac{\Lambda }{\bar{X}+\Lambda }\frac{\partial {\bar{U}}_{2}}{\partial \bar{S}}+\frac{{\bar{U}}_{1}}{\bar{X}+\Lambda }\right)}^{2}\right\}+{\left(\frac{\partial {\bar{U}}_{2}}{\partial \bar{X}}+\frac{\Lambda }{\bar{X}+\Lambda }\frac{\partial {\bar{U}}_{1}}{\partial \bar{S}}-\frac{{\bar{U}}_{2}}{\bar{X}+\Lambda }\right)}^{2}\right)\;+\frac{\sigma_{\mathrm{hnf}}B_{0}^{2}\Lambda^{2}}{{\left(\bar{X}+\Lambda \right)}^{2}+{\left(m\Lambda \right)}^{2}}U_{2}^{2}.$$
with $\mu_{hnf}$ (viscosity), ${\left(\rho C_{p}\right)}_{hnf}$ (heating capacity), $\rho_{hnf}$ (density), $\gamma_{1}$ (Casson factor) and $K_{hnf}$ (thermal conductivity). Table 1 is organized in order to present the flow properties of the hybrid model.

Changing the problem from fixed to waves frame by introducing the following transportations:(12)$$\bar{s}=\bar{S}-c\bar{t},\bar{X}=\bar{x},{\bar{u}}_{1}={\bar{U}}_{1},{\bar{u}}_{2}={\bar{U}}_{2}-c,\bar{P}=\bar{p}$$

Following the above transformations, Equations (9)–(12) become:(13)$$\frac{\partial }{\partial \bar{x}}\left\{\left(\bar{x}+\Lambda \right){\bar{u}}_{1}\right\}+\Lambda \frac{\partial {\bar{u}}_{2}}{\partial \bar{s}}=0,$$
(14)$$\rho_{hnf}\left(-c\frac{\partial {\bar{u}}_{1}}{\partial \bar{s}}+u_{1}\frac{\partial {\bar{u}}_{1}}{\partial \bar{x}}+\frac{\left({\bar{u}}_{2}+c\right)\Lambda }{\bar{X}+\Lambda }\frac{\partial {\bar{u}}_{1}}{\partial \bar{s}}-\frac{{\left({\bar{u}}_{2}+c\right)}^{2}}{\bar{x}+\Lambda }\right)=\;-\frac{\partial \bar{p}}{\partial \bar{x}}+\mu_{hnf}\left(1+\frac{1}{\gamma_{1}}\right)\left(\frac{1}{\bar{x}+\Lambda }\frac{\partial }{\partial \bar{x}}\left(\left(\bar{x}+\Lambda \right)\frac{\partial {\bar{u}}_{1}}{\partial \bar{x}}\right)+{\left(\frac{\Lambda }{\bar{x}+\Lambda }\right)}^{2}\frac{\partial^{2}{\bar{u}}_{1}}{\partial {\bar{s}}^{2}}-\frac{2\Lambda }{{\left(\bar{x}+\Lambda \right)}^{2}}\frac{\partial {\bar{u}}_{2}}{\partial \bar{s}}-\frac{u_{1}}{{\left(\bar{x}+\Lambda \right)}^{2}}\right)$$
(15)$$\rho_{hnf}\left(-c\frac{\partial {\bar{u}}_{2}}{\partial \bar{s}}+{\bar{u}}_{1}\frac{\partial {\bar{u}}_{2}}{\partial \bar{x}}+\frac{\left({\bar{u}}_{2}+c\right)\Lambda }{\bar{x}+\Lambda }\frac{\partial {\bar{u}}_{2}}{\partial \bar{s}}-\frac{{\bar{u}}_{1}\left({\bar{u}}_{2}+c\right)}{\bar{x}+\Lambda }\right)=\;-\frac{\Lambda }{\bar{x}+\Lambda }\frac{\partial \bar{p}}{\partial \bar{s}}+\mu_{hnf}\left(1+\frac{1}{\gamma_{1}}\right)\left(\frac{1}{\bar{x}+\Lambda }\frac{\partial }{\partial \bar{x}}\left(\left(\bar{x}+\Lambda \right)\frac{\partial {\bar{u}}_{2}}{\partial \bar{x}}\right)+{\left(\frac{\Lambda }{\bar{x}+\Lambda }\right)}^{2}\frac{\partial^{2}{\bar{u}}_{2}}{\partial {\bar{s}}^{2}}+\frac{2\Lambda }{{\left(\bar{x}+\Lambda \right)}^{2}}\frac{\partial {\bar{u}}_{1}}{\partial \bar{s}}-\frac{{\bar{u}}_{2}+c}{{\left(\bar{x}+\Lambda \right)}^{2}}\right)\;-\frac{\sigma_{hnf}B_{0}^{2}\Lambda^{2}}{{\left(\bar{x}+\Lambda \right)}^{2}}\left({\bar{u}}_{2}+c\right)\left(\frac{1}{1+{\left(\frac{m\Lambda }{\bar{x}+\Lambda }\right)}^{2}}\right)$$
(16)$${\left(\rho C_{p}\right)}_{hnf}\left({\bar{u}}_{1}\frac{\partial \bar{T}}{\partial \bar{x}}+\frac{\left({\bar{u}}_{2}+c\right)\Lambda }{\bar{x}+\Lambda }\frac{\partial \bar{T}}{\partial \bar{s}}\right)=\;K_{hnf}\left(\frac{1}{\bar{x}+\Lambda }\frac{\partial }{\partial \bar{x}}\left(\left(\bar{x}+\Lambda \right)\frac{\partial \bar{T}}{\partial \bar{x}}\right)+{\left(\frac{\Lambda }{\bar{x}+\Lambda }\right)}^{2}\frac{\partial^{2}\bar{T}}{\partial {\bar{s}}^{2}}\right)\;+\mu_{hnf}\left(1+\frac{1}{\gamma_{1}}\right)\left(2\left\{{\left(\frac{\partial {\bar{u}}_{1}}{\partial \bar{x}}\right)}^{2}+{\left(\frac{\Lambda }{\bar{X}+\Lambda }\frac{\partial \left({\bar{u}}_{2}\right)}{\partial \bar{s}}+\frac{{\bar{u}}_{1}}{\bar{x}+\Lambda }\right)}^{2}\right\}+{\left(\frac{\partial \left({\bar{u}}_{2}\right)}{\partial \bar{x}}+\frac{\Lambda }{\bar{x}+\Lambda }\frac{\partial {\bar{u}}_{1}}{\partial \bar{s}}-\frac{\left({\bar{u}}_{2}+c\right)}{\bar{x}+\Lambda }\right)}^{2}\right)\;+\frac{\sigma_{\mathrm{hnf}}B_{0}^{2}\Lambda^{2}}{{\left(\bar{x}+\Lambda \right)}^{2}+{\left(m\Lambda \right)}^{2}}{\left({\bar{u}}_{2}+c\right)}^{2},$$

Now we use the dimensionless variables [34,35,36]:$$s=\frac{2\pi \bar{s}}{\beta },\;\eta =\frac{\bar{x}}{b},\;u_{1}=\frac{{\bar{u}}_{1}}{c},\;u_{2}=\frac{{\bar{u}}_{2}}{c},\;\delta =\frac{2\pi b}{\beta },\;h=\frac{\bar{H}}{b},\;Re=\frac{\rho_{f}cb}{\mu_{f}},\;k=\frac{\Lambda }{b},\;p=\frac{2\pi b^{2}}{\beta \mu_{f}c}\bar{p},.\;$$
$$\Phi_{i}=\frac{{\bar{\Phi }}_{i}}{b},i=1,2,m_{1}=\frac{\beta \bar{M}}{b},\;Ha=\sqrt{\frac{\sigma_{f}}{\mu_{f}}}B_{0}b,\;\theta =\frac{\bar{T}-T_{1}}{T_{2}-T_{1}},\;Ec=\frac{c^{2}}{C_{f}\left(T_{2}-T_{1}\right)},\;Pr=\frac{\mu_{f}C_{f}}{K_{f}},.\;$$
Br=PrEc,
with η (radial direction component), $u_{1}$ (radial velocity), s (axial component), $u_{2}$ (axial velocity), Re (Reynolds constant), Pr (Prandtl constant), k (dimensionless curvature), Ha (Hartmann number), δ (wave number), M (non-uniform parameter), Ec (Eckert number) and Br (Brinkman number). Using stream functions defining as $u_{1}=\delta \left(\Lambda /\eta +\Lambda \right)\partial \psi /\partial s\;$ and $\;u_{2}=-\partial \psi /\partial \eta \;$ and implementing the smaller Reynolds assumptions, the rest of equations take the form [34,35,36]:(17)$$\frac{\partial p}{\partial \eta }=0.$$
(18)$$\frac{\partial p}{\partial s}=\frac{1}{k\left(k+\eta \right)}\frac{1}{{\left(1-\alpha_{1}-\alpha_{2}\right)}^{2.5}}\left(1+\frac{1}{\gamma_{1}}\right)\frac{\partial }{\partial \eta }\left(\left(\eta +k\right)\left(1-\frac{\partial \psi }{\partial \eta }\right)+{\left(\eta +k\right)}^{2}\frac{\partial^{2}\psi }{\partial \eta^{2}}\right)-C_{1}\frac{Ha^{2}k^{2}}{{\left(\;\eta +k\right)}^{2}+{\left(mk\right)}^{2}}\left(1-\frac{\partial \psi }{\partial \eta }\right),$$
(19)$$C_{2}\left(\frac{\partial^{2}\theta }{\partial \eta^{2}}+\frac{1}{\eta +k}\frac{\partial \theta }{\partial \eta }\right)+\frac{Br}{{\left(1-\alpha_{1}-\alpha_{2}\right)}^{2.5}}\left(1+\frac{1}{\gamma_{1}}\right){\left(\frac{\partial^{2}\psi }{\partial \eta^{2}}+\frac{1}{\eta +k}\left(1-\frac{\partial \psi }{\partial \eta }\right)\right)}^{2}+\frac{k^{2}BrHa^{2}C_{1}}{{\left(\;\eta +k\right)}^{2}+{\left(mk\right)}^{2}}{\left(1-\frac{\partial \psi }{\partial \eta }\right)}^{2}=0.$$
with the copper nanoparticles’ volume fraction $(\alpha_{1}$) and ferro nanomaterials volume fraction of $\alpha_{2}$. Moreover $C_{1}=\frac{\sigma_{s_{1}}+2\sigma_{bf}-2\alpha_{2}\left(\sigma_{bf}-\sigma_{s_{1}}\right)}{\sigma_{s_{1}}+2\sigma_{bf}+\alpha_{2}\left(\sigma_{bf}-\sigma_{s_{1}}\right)},$ where $\sigma_{bf}=\frac{\sigma_{s_{2}}+2\sigma_{f}-2\alpha_{1}\left(\sigma_{f}-\sigma_{s_{2}}\right)}{\sigma_{s_{2}}+2\sigma_{f}+\alpha_{1}\left(\sigma_{f}-\sigma_{s_{2}}\right)}\sigma_{f}$ and $C_{2}=\frac{K_{hnf}}{K_{f}}=\frac{\frac{\alpha_{1}K_{s_{1}}+\alpha_{2}K_{s_{2}}}{\alpha_{1}+\alpha_{2}}+2K_{f}-2K_{f}\left(\alpha_{1}+\alpha_{2}\right)+2\left(\alpha_{1}K_{s_{1}}+\alpha_{2}K_{s_{2}}\right)}{\frac{\alpha_{1}K_{s_{1}}+\alpha_{2}K_{s_{2}}}{\alpha_{1}+\alpha_{2}}+2K_{f}+K\left(\alpha_{1}+\alpha_{2}\right)-\left(\alpha_{1}K_{s_{1}}+\alpha_{2}K_{s_{2}}\right)}$.

The boundary conditions in terms of stream function [34,35,36]:(20)$$\psi =-\frac{q}{2},\frac{\partial \psi }{\partial \eta }+\beta_{1}\left(1+\frac{1}{\gamma_{1}}\right)\frac{1}{{\left(1-\alpha_{1}-\alpha_{2}\right)}^{2.5}}\left(-\frac{\partial^{2}\psi }{\partial \eta^{2}}-\frac{1}{\eta +k}\left(1-\frac{\partial \psi }{\partial \eta }\right)\right)=1,\theta =1$$
(21)$$\mathrm{at}\;\eta =h_{2}=1+m_{1}s+\Phi_{1}\sin\left(\sigma s\right)+\Phi_{2}\sin\left(\omega s\right)$$
(22)$$\psi =\frac{q}{2},\frac{\partial \psi }{\partial \eta }-\beta_{1}\left(1+\frac{1}{\gamma_{1}}\right)\frac{1}{{\left(1-\alpha_{1}-\alpha_{2}\right)}^{2.5}}\left(-\frac{\partial^{2}\psi }{\partial \eta^{2}}-\frac{1}{\eta +k}\left(1-\frac{\partial \psi }{\partial \eta }\right)\right)=1,\theta =0$$
(23)$$\mathrm{at}\;\;\eta =h_{1}=-1-m_{1}s-\Phi_{1}\sin\left(\sigma (s+\epsilon \right)-\Phi_{2}\sin\left(\omega \left(s+\epsilon \right)\right)$$

For mean flow:(24)$$\Omega_{1}=q+2$$

Defining q as:(25)$$q=\int_{-h}^{h}\frac{\partial \psi }{\partial \eta }d\eta ,$$

The fluctuation in pressure rise, wall shear force and heating transfer coefficient near the lower surface wall and upper surface is:(26)$$\Delta p=\int_{0}^{2\pi }\frac{dp}{dx}dx$$
(27)$$C_{f}=-{\frac{\partial h_{1}}{\partial s}\frac{\partial^{2}\psi }{\partial \eta^{2}}|}_{h_{1}}$$
(28)$$Z_{1}={\frac{\partial h_{2}}{\partial s}\frac{\partial \theta }{\partial \eta }|}_{h_{2}}$$

## 3. Solution of the Problem

After eliminating pressure for Equations (17) and (18), the shooting method with built-in ND solver is applied. This scheme is well-known and is not presented in detail here. Table 2 interprets the numerical values of density, thermal conductivity, specific heat and electrical conductivity. 

## 4. Discussion

This section claims some physical importance of the parameters for different flow regimes. For this task, the values are parameters which are defined as $\sigma =1,\;m_{1}=0.2,\;\omega =1\;$ and ϵ=π.

### 4.1. Axial Velocity Profile

Figure 2 is the graphical representation of the axial velocity against the several involved parameters for non-similar values of flow rate q, i.e., q=0.1 and q=−5.0. The velocity profile is plotted and examined under the influence of both the slip and non-Newtonian rheology of the nanofluid. From Figure 2a, it is clear that the velocity of the hybrid nanofluid declines at the central line of the channel by boosting the solid volume fraction of Cu nanoparticles and near the walls of the channel the velocity rises; this behavior is same for several values of q. Physically, these results show that the solid volume fraction plays a dynamic role in the transport of blood near the lubricated walls of curved micro-vascular conduits in the presence of hybrid nanoparticles. The solid volume fraction of copper nanoparticles can also play a key role in the regulation of blood in diseased arteries with lubricated walls. A similar trend is observed in Figure 2b for the axial velocity by boosting the solid volume fraction of Fe_3_O_4_ nanoparticles for both q=0.1 and q=−5.0. This indicates that due to the interaction of the nanoparticles, which control the thermal transport near the micro channel, there is a decrease in velocity in the center of the channel, and due to the slip features at the walls, the velocity rises. Figure 2c is plotted to analyze the change in the axial velocity component due to increasing fluctuation of the Hartmann number. The reduced results are observed in the upper regime when enhancing values are being assigned to the Hartmann number. However, a rising velocity in the lower regime is observed for the same Hartmann constant variation. The change in the Hartmann number against the axial velocity is described via Figure 2d. Reduced velocity in the core of the channel is observed. As a result, a depressive velocity trend occurs and the velocity fluctuates toward the upper regime of the channel. The reduction in velocity flow is due to the application of magnetic force which results in a resistive Lorentz force. The control of velocity due to $m_{1}$ has been noted in Figure 2e. The enhanced change in hybrid nanofluid movement against $m_{1}$ in the core regime is noticed. Moreover, the channel diameter begins to decrease when larger measurements are assigned to $m_{1}$. Figure 2f presents important observations of the change in velocity caused by the impact of curvature k. The results are further observed for an infinite range of curvature. The response of velocity is similar for non-similar values of flow rate q.

### 4.2. Temperature Profile

The results described in Figure 3a present the trends of temperature profile θ(η) due to the Casson fluid parameter for Cu/blood nanofluid and hybrid nanofluid [34,35,36]. The noted observations reveal that both Cu/blood nanofluid and hybrid nanofluid are a decreasing function of the Casson parameter. As the Casson parameter increases, the behavior of the fluid approach that of viscous fluid, so for viscous fluid the temperature is minimal and for the Casson fluid the temperature is maximal. As the non-Newtonian character is added, the resistance between the layers of fluid rises, increasing the internal kinetic energy of the colloidal suspension. As a result, the enhancing temperature rate is deduced [34,35,36]. The same nature of temperature results from enrolling the change in the velocity slip constant (Figure 2b). The Hall current reduces the role of Lorentz force as seen in Figure 2c. The graphical observations in Figure 2d demonstrate an increase in the thermal rate caused by an increasing Hartmann number. An improvement in the thermal performance of the copper–blood suspension is observed due to Lorentz force. Similarly, an improved heat transfer phenomenon is noted for ferro-copper nanoparticles. Such results may have important implications for applications within ferromagnetic materials and industrial processes. An increase in temperature is visualized when the non-uniformity of the channel increases (Figure 2e). Therefore, the non-uniformity of the surface geometry plays an important role in the thermal transmission of various processes. Figure 2f demonstrates a low temperature rate for an increasing curvature constant. 

### 4.3. Trapping Phenomena

In many situations the boluses of fluid are generated in the flow regimes which are enclosed by streamlines; this occurrence is referred to as trapping phenomena. This trapping is an important rheological feature related to the dynamics of the bolus under various circumstances. A wavy streamlined shape is produced in the vicinity of the walls on both halves of the channel due to the complex wave pattern on the wall. Furthermore, the flow structure of the trapping phenomena can be discussed and predicted due to the non-uniformity of the flow patterns. Figure 4, Figure 5, Figure 6 and Figure 7 demonstrate the significance of $\alpha_{1}$, $\alpha_{2}$, Ha, m and $m_{1}$. The nature of the bolus is noted for the parameters defined in Figure 4. A smaller bolus size is exhibited, which disappears in the upper channel regime when the volume fraction rate increases. From Figure 4b, Figure 5b, Figure 6b and Figure 7b, larger bolus size in associated to the increasing volume fraction rate. Such a trend is due to the addition of solid nanoparticles to the base liquid. From Figure 5, it can be observed that the concentration of the bolus plays no role on either of the channel surfaces. The appearance of the bolus in the channel is observed without the participation of the magnetic phenomenon. Actually, the fluid trapped towards the wall of the channel and the flow patterns near the walls develop under strong magnetic force. Figure 6 shows the effect of the onset of the Hall factor on the trapping phenomenon; the opposite effect is noted for the Hall parameter. From Figure 7, the bolus is seen to disappear inside the channel when movement is observed in the uniform and non-uniform channel regimes.

### 4.4. Pressure Rise

Figure 8a,b is presented to visualize the change in the pumping phenomenon against the nanomaterials’ volume fraction. Owing to the larger nanomaterial concentration, the increment in pressure is observed, while the contrary trend is noticed in the co-pumping regime. Moreover, slightly larger pressure distribution is results from copper nanoparticles. Such results are significant for roller pumping design due to curvature. Figure 8c,d demonstrates the phenomenon of pressure change for the Hartmann number and the Hall factor. The enhancing pressure gradient is observed in the pumping core while the decrease is noted in the co-pumping zone.

### 4.5. Heat Transfer Coefficient and Wall Shear Force

The numerical claims are listed in Table 3 to observe the change in heat transfer coefficient for copper nanoparticles $\left(\alpha_{1}=0.1,\alpha_{2}=0.0\right)$ and hybrid nanofluid $\left(\alpha_{1}=0.1,\alpha_{2}=0.2\right)$. The analysis is investigated for the hybrid nanofluid model and the blood/copper nanofluid suspension. An increase in the heat transfer coefficient is noted upon enlarging the Hartmann factor. However, the larger transmission of energy is noted for the copper–blood decomposition. The role of the slip factor controls the heating phenomenon for both models. Observations of the decrement when varying the Hall parameter yield results. For the blood–copper model, the heat transfer factor increases for the Brinkman number as well as for the hybrid nanofluid, both with the no-slip mechanism and by entertaining the slip factor. Moreover, the increasing trend in the curved channel is noted. From Table 4, the numerical trend of the wall shear force increases for the Hartmann constant for all models. However, the low wall shear results are determined for the hybrid model. The Hall current factor and curvature constant present decreasing changes for the wall shear quantity.

## 5. Conclusions

The peristatic pumping-based blood flow with applications of hybrid nanofluid is studied in a curved channel. The determination of heat transfer is observed in ferro nanoparticles and copper nonmaterial. Additionally, the importance of the Hall factor contributes to make the model comprehensive. The significance of various flow parameters for velocity, heat transfer coefficients and pumping phenomena are visualized. The mains results are concluded as:❖The declining change in velocity associated with the larger Lorentz force is exhibited.❖Upon enhancing the Hall parameter, the role of magnetic force is controlled.❖The change in the non-uniformity of a curved surface and the increment in the rate of velocity are observed.❖A reduction in temperature results from a larger nanoparticle volume fraction.❖The heat transfer is increased for the curved configuration while lower results are noted for the planner channel.❖In both regimes of the symmetric channel, the disappearance of bolus trapping due to magnetic force is noted.❖A progressive skin friction for hybrid nanofluid at low scales is given for Lorentz force.

## Figures and Tables

**Figure 1 micromachines-13-01415-f001:**
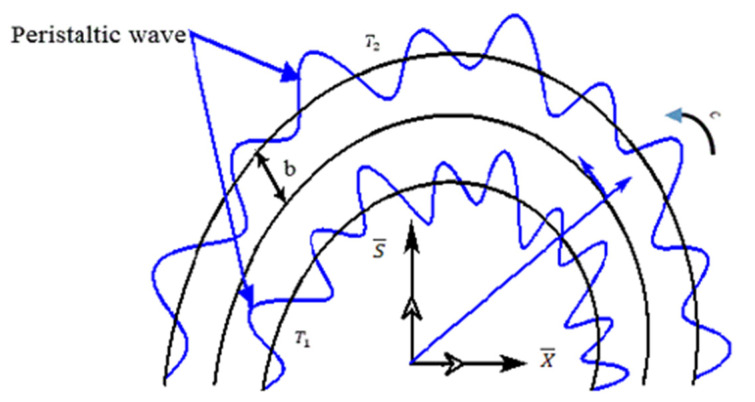
Flow illustration of model.

**Figure 2 micromachines-13-01415-f002:**
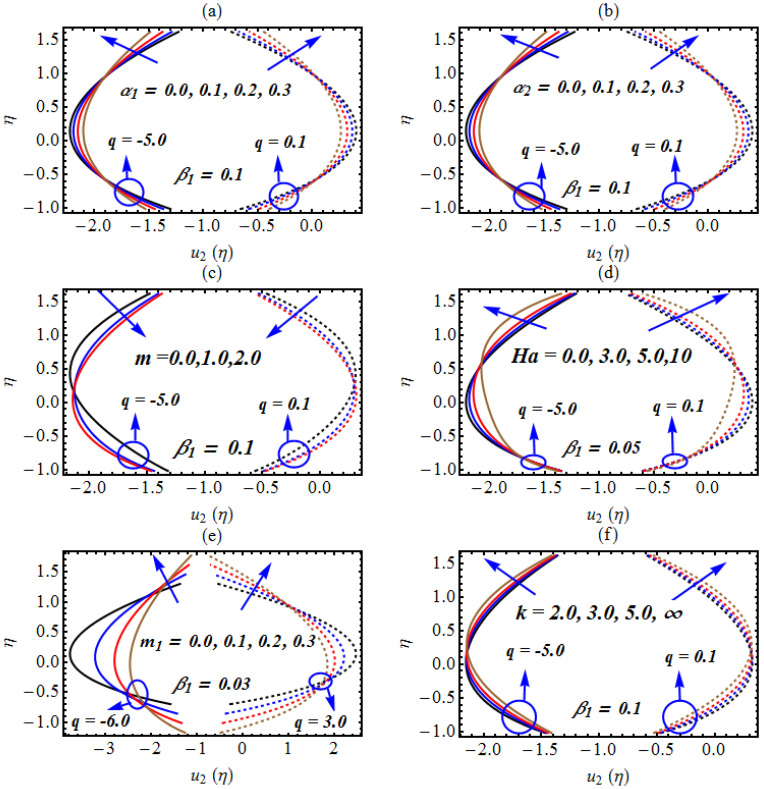
(**a**–**f**): (**a**) Change in *u*_2_(*η*) for $\alpha_{1}$, (**b**) change in *u*_2_(*η*) for $\alpha_{2}$, (**c**) change in *u*_2_(*η*) for m, (**d**) change in *u*_2_(*η*) for Ha, (**e**) change in *u*_2_(*η*) for *m*_1_, (**f**) change in *u*_2_(*η*) for *m*_1_/.

**Figure 3 micromachines-13-01415-f003:**
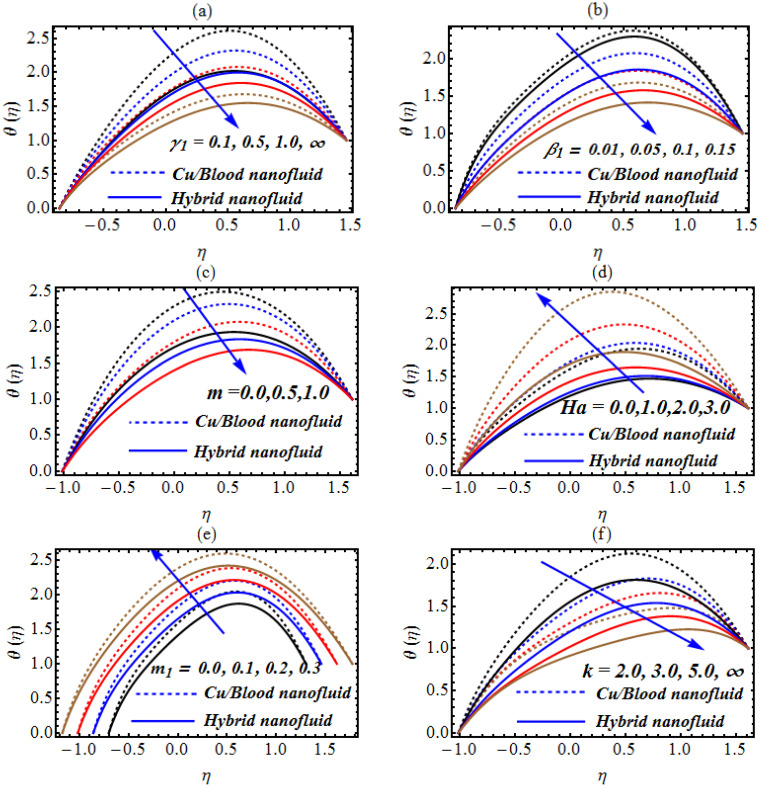
(**a**–**f**): (**a**) Change in θ for $\gamma_{1}$ (**b**) change in θ for $\beta_{1}$, (**c**) change in θ for m, (**d**) change in θ for Ha, (**e**) change in θ for $m_{1}$ and (**f**) change in θ for k.

**Figure 4 micromachines-13-01415-f004:**
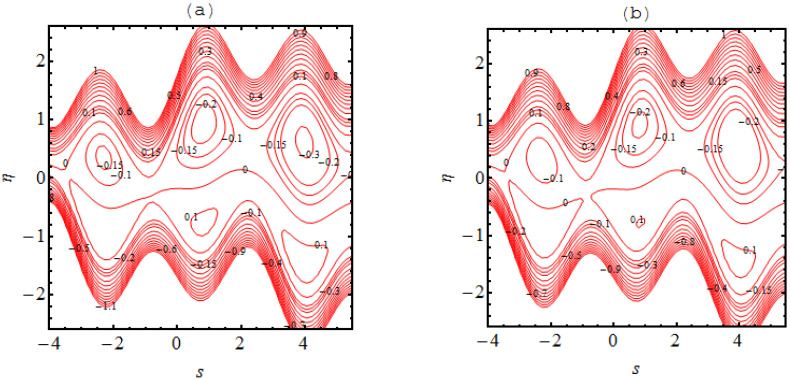
Stream lines for (**a**) $\alpha_{1}=0.0$ and (**b**) $\alpha_{1}=0.1$ the other parameters are $Ha=3.0,\;k=3.0,\;m=2.0,\;q=-0.1,\gamma_{1}=1.0,\beta_{1}=0.1$.

**Figure 5 micromachines-13-01415-f005:**
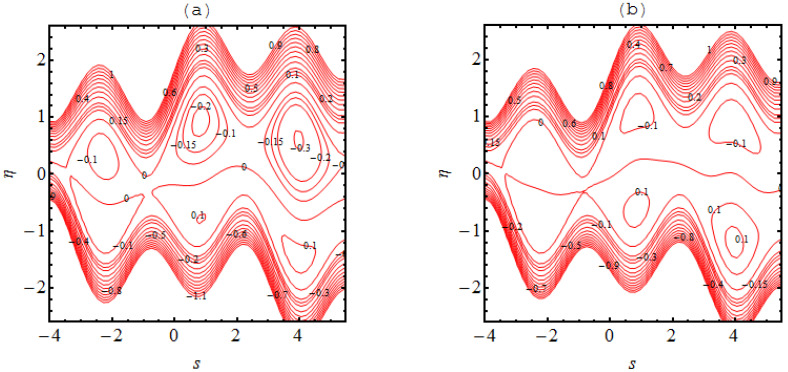
Stream lines for (**a**) Ha=0.0 and (**b**) Ha=3.0 the other parameters are $\alpha_{1}=0.05,\;k=3.0,\;m=2.0,\;q=-0.1$ and $\alpha_{2}=0.1$.

**Figure 6 micromachines-13-01415-f006:**
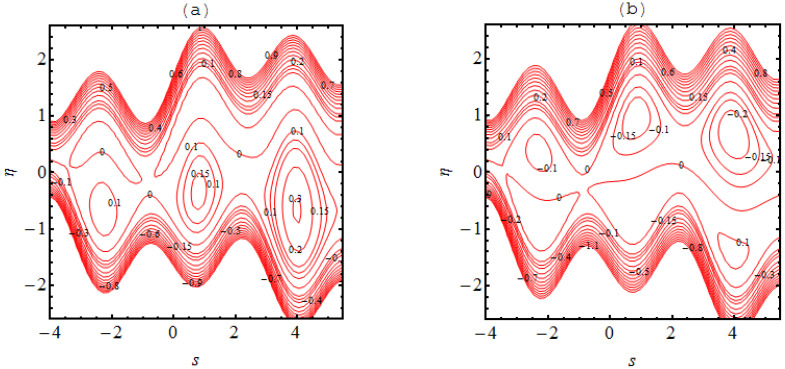
Stream lines for (**a**) m=0.0 and (**b**) m=2.0 the other parameters are $\alpha_{1}=0.05,\;k=3.0,\;Ha=3.0,\;q=-0.1$ and $\;\alpha_{2}=0.1$.

**Figure 7 micromachines-13-01415-f007:**
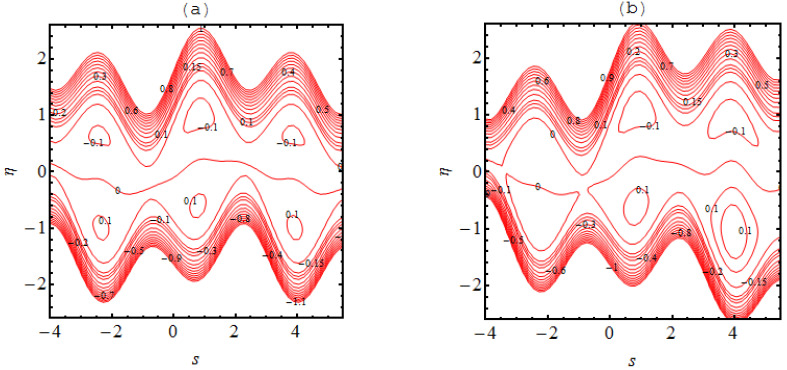
Stream lines for (**a**) $m_{1}=0.0$ and (**b**) $m_{1}=0.1\;$ where the other parameters are $\alpha_{2}=0.1,\;k=3.0,\;Ha=3.0,\;q=-0.1,m=1.0$ and $\alpha_{1}=0.05$.

**Figure 8 micromachines-13-01415-f008:**
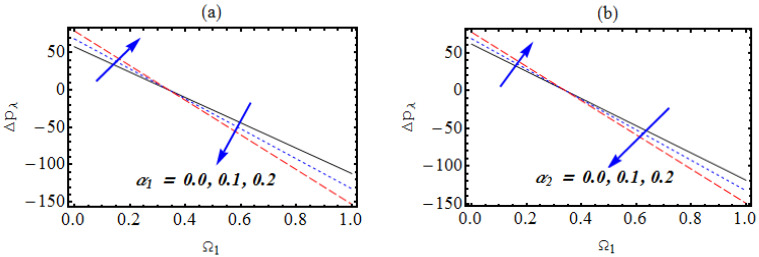
(**a**–**d**): (**a**) Change in pressure due to $\alpha_{1},$ (**b**) change in pressure due to $\alpha_{2},$ (**c**) change in pressure due to Ha (**d**) change in pressure due to m.

**Table 1 micromachines-13-01415-t001:** Hybrid nanofluid different consequences with mathematical forms [5].

Density	$\rho_{hnf}=\left(1-\alpha_{1}-\alpha_{2}\right)\rho_{f}+\alpha_{1}\rho_{s_{1}}+\alpha_{2}\rho_{s_{2}}$.
Viscosity	$\mu_{hnf}=\frac{\mu_{f}}{{\left(1-\alpha_{1}-\alpha_{2}\right)}^{2.5}}$
Effective heat capacity	${\left(\rho C_{p}\right)}_{hnf}=\left(1-\alpha_{1}-\alpha_{2}\right)\rho_{f}{\left(C_{p}\right)}_{f}+\alpha_{1}\rho_{s_{1}}{\left(C_{p}\right)}_{s_{1}}+\alpha_{2}\rho_{s_{2}}{\left(C_{p}\right)}_{s_{2}}$
Thermal conductivity	$\frac{K_{hnf}}{K_{f}}=\frac{\frac{\alpha_{1}K_{s_{1}}+\alpha_{2}K_{s_{2}}}{\alpha_{1}+\alpha_{2}}+2K_{f}-2K_{f}\left(\alpha_{1}+\alpha_{2}\right)+2\left(\alpha_{1}K_{s_{1}}+\alpha_{2}K_{s_{2}}\right)}{\frac{\alpha_{1}K_{s_{1}}+\alpha_{2}K_{s_{2}}}{\alpha_{1}+\alpha_{2}}+2K_{f}+K\left(\alpha_{1}+\alpha_{2}\right)-\left(\alpha_{1}K_{s_{1}}+\alpha_{2}K_{s_{2}}\right)}$
Electric conductivity	$\frac{\sigma_{hnf}}{\sigma_{bf}}=\frac{\sigma_{s_{1}}+2\sigma_{bf}-2\alpha_{2}\left(\sigma_{bf}-\sigma_{s_{1}}\right)}{\sigma_{s_{1}}+2\sigma_{bf}+\alpha_{2}\left(\sigma_{bf}-\sigma_{s_{1}}\right)}$ $\mathrm{where}\;\sigma_{bf}=\frac{\sigma_{s_{2}}+2\sigma_{f}-2\alpha_{1}\left(\sigma_{f}-\sigma_{s_{2}}\right)}{\sigma_{s_{2}}+2\sigma_{f}+\alpha_{1}\left(\sigma_{f}-\sigma_{s_{2}}\right)}\sigma_{f}$

**Table 2 micromachines-13-01415-t002:** The properties of ferro nanoparticles, copper and human blood [5,16].

Material	Cu	Blood	Fe_3_O_4_
$\rho \;\left(\mathrm{kg}/m^{3}\right)$	8933	1063	5200
K (W/mk)	401	0.492	6
C (J/kgK)	385	3594	670
σ (S/m)	$5.96\times 10^{7}$	0.8	25,000

**Table 3 micromachines-13-01415-t003:** Variation of heat transfer coefficient with $\gamma_{1}=1.0$ and q=−0.2.

Ha	m	Br	k	$\beta_{1}=0.0$		$\beta_{1}=0.1$	
				Cu Nanofluid	Hybrid Nanofluid	Cu Nanofluid	Hybrid Nanofluid
0.0	1.0	1.0	3.0	1.946431	1.803762	1.313924	1.305188
1.0				1.947308	1.804217	1.314652	1.305564
2.0				1.949939	1.805583	1.316832	1.306690
	0.0			1.952836	1.807087	1.319297	1.307965
	1.0			1.949939	1.805583	1.316832	1.306690
	2.0			1.947920	1.804535	1.315149	1.305820
		0.0		1.283249	1.283249	1.283249	1.283249
		2.0		2.56784366	2.28255142	1.30959600	1.290532
		4.0		3.85243826	3.28185379	1.33594274	1.29781534
			2.5	1.95869391	1.81494916	1.32416001	1.31492136
			5.0	1.92554637	1.78290025	1.29642262	1.28689076
			∞	1.88993547	1.74844033	1.26657588	1.25666942

**Table 4 micromachines-13-01415-t004:** Change in skin friction with $\gamma_{1}=1.0$ and q=−0.2.

Ha	m	k	$\beta_{1}=0.0$		$\beta_{1}=0.1$	
			Cu Nanofluid	Hybrid Nanofluid	Cu Nanofluid	Hybrid Nanofluid
0.0	1.0	3.0	1.64793574	1.647935744	0.23902529	0.144332354
1.0			1.64825439	1.64814824	0.23902712	0.14433235
2.0			1.64921003	1.64878559	0.23903264	0.14433235
	0.0		1.65017632	1.64943017	0.23899158	0.14431076
	1.0		1.64921003	1.64878559	0.23903264	0.14433236
	2.0		1.64849017	1.64830548	0.23903597	0.14433582
		2.5	1.65630099	1.65612141	0.24164339	0.14724496
		5.0	1.63224142	1.63204609	0.23360486	0.13824367
		∞	1.60630293	1.60609039	0.22491558	0.12840275

## Data Availability

Data regarding current research is mentioned in the manuscript.

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
