# Peer review of "Heat Transport Exploration for Hybrid Nanoparticle (Cu, Fe3O4)—Based Blood Flow via Tapered Complex Wavy Curved Channel with Slip Features"

_micromachines, 2022, doi:10.3390/mi13091415_

Round 1

Reviewer 1 Report

Micromachines - 1877929

Comments from Reviewer

This paper seems to be interesting but there are several flaws that must be addressed to significantly improve the paper in its present form as given in the comments below. In this regard, I do not recommend this paper to be published and major revision is required for the paper in its present form.

1.         Check font style and size. They are not consistent.

2.         In lines 80-81 “….nanoparticles particles” sounds strange. Is this a typo?

3.         What is the definition of hybrid nanofluid? How is it better than nanofluid is technicality? Please explain in detail.

4.         The motivation of the study is not clear. Authors should explain specifically what is different from those studies cited for [34-36]. What is the novelty of this study compared to those from [34-36]?

5.         Equations from lines 204 to 206 do not have equation numbers. Please check and rectify.

6.         Table 2 is not mentioned in Section 2. What is the purpose of showing Table 2? How were that info in Table 2 obtained?

7.         Section 3 is too short and seems redundant. Those should be mentioned in Section 2 instead, where Eqs. 17 and 18 are mentioned.

8.         What is the meaning of ND? It would be useful to add in a list of nomenclature and list of symbols so readers can refer to those unknown abbreviations. This will make it much easier for readers to understand the whole technical terms used.

9.         The hybrid nanofluid model developed in this study is not validated against the experiment and there are assumptions taken into consideration for the model, therefore the accuracy of results presented are questionable.

10.       Please indicate in Figs. 2 to 9 (Except Fig. 3) which one is Cu and which one is without Cu in the graphs. Also indicate what are those colored dashed lines for each graph. 

11.       Is this study related to Cu or Cu, Fe3O4? The title mentioned Cu, Fe3)4 but the results stated Cu only.

12.       It is difficult to understand the trend in Tables 3 and 4 as there are too many numbers. It would be better to illustrate the variations in graphical presentation.

13.       The novelty is not clear. Please emphasize the novelty and contribution of the present study in the conclusion.

14.       Please add recommendations for future work and highlight the new knowledge of this study at the end of the conclusion.

15.       There are a few typos/grammatical errors spotted. Please check through the paper and correct those errors.

Author Response

Response to the Reviewer #1

The authors are thankful to the anonymous reviewer for useful comments. These suggestions really enhance the quality of the manuscript. The detail response to the reviewer’s suggestions is as follows.

Micromachines - 1877929

Comments from Reviewer

This paper seems to be interesting but there are several flaws that must be addressed to significantly improve the paper in its present form as given in the comments below. In this regard, I do not recommend this paper to be published and major revision is required for the paper in its present form.

  1. Check font style and size. They are not consistent.

Response: Now fint size is checked and corrected.

  1. In lines 80-81 “….nanoparticles particles” sounds strange. Is this a typo?

Response: This typo error is now corrected.

  1. What is the definition of hybrid nanofluid? How is it better than nanofluid is technicality? Please explain in detail.

 Response: The hybrid nanofluid is the superior category of nanofluids which reflects the improved thermal consequences of based materials with suspension of two different nanoparticles. The motivations for being the observing the thermal mechanism of base fluid via hybrid nanofluid are due to enhanced thermal mechanism. The reflection of hybrid nanofluid model have more fascinating behavior rather than nanofluids. Special applications of hybrid nanofluids are observed the energy production, manufacturing systems, solar applications, heating devices, countering the thermal rate of engine, extrusion processes etc.  The class of hybrid nanofluids are achieved after compassion of base material in more than one tiny nanoparticles. This explanation is added in the introduction section.

  1. The motivation of the study is not clear. Authors should explain specifically what is different from those studies cited for [34-36]. What is the novelty of this study compared to those from [34-36]?

Response: Motivations of current work are now improved. The difference between current work and refs. [34-36] is clearly presented.

  1. Equations from lines 204 to 206 do not have equation numbers. Please check and rectify.

 Response: Now proper equation numeri is assigned.

  1. Table 2 is not mentioned in Section 2. What is the purpose of showing Table 2? How were that info in Table 2 obtained?

Response: Now clearly mentioned. This table preset thermal characteristics of hybrid nanofluids.

  1. Section 3 is too short and seems redundant. Those should be mentioned in Section 2 instead, where Eqs. 17 and 18 are mentioned.

Response: Now improved.

  1. What is the meaning of ND? It would be useful to add in a list of nomenclature and list of symbols so readers can refer to those unknown abbreviations. This will make it much easier for readers to understand the whole technical terms used.

Response: The ND solver is a built in numerical technique.

  1. The hybrid nanofluid model developed in this study is not validated against the experiment and there are assumptions taken into consideration for the model, therefore the accuracy of results presented are questionable.

Response: This this investigation is based on some theoretical flow assumptions. So current model is not validated with experimental data.

  1. Please indicate in Figs. 2 to 9 (Except Fig. 3) which one is Cu and which one is without Cu in the graphs. Also indicate what are those colored dashed lines for each graph. 

 Response: Now explanation is added.

  1. Is this study related to Cu or Cu, Fe3O4? The title mentioned Cu, Fe3)4 but the results stated Cu only.

 Response: The Cu and Fe3O4 nanoparticles are used express the hybrid nanofluid properties. The results for Cu are only for representation of simple nanofluid.

  1. It is difficult to understand the trend in Tables 3 and 4 as there are too many numbers. It would be better to illustrate the variations in graphical presentation.

 Response: Now improved.

  1. The novelty is not clear. Please emphasize the novelty and contribution of the present study in the conclusion.

 Response: Motivations of current work are now improved. The difference between current work and already performed studies is clearly presented.

  1. Please add recommendations for future work and highlight the new knowledge of this study at the end of the conclusion.

 Response: Now done.

  1. There are a few typos/grammatical errors spotted. Please check through the paper and correct those errors.

Response: Some typo errors are now corrected.

Reviewer 2 Report

The report presents the outcome of a study on blood conveying Iron(II,III) oxide and Copper nanoparticles through tapered complex wavy curved channel with slip features. The contribution of the report to the body of knowledge is significant. Noteworthy that the objectives of the research study are within the scope of Micromachines. But, the present form of the report needs revision using the comments and the questions/observations listed below   Q1. The present form of the title is not acceptable. Revise or consider: Exploration of blood conveying Iron(II,III) oxide and Copper nanoparticles through tapered complex wavy curved channel with slip features   Q2. Abstract, line 28, it was written, “In the human cardiovascular system, the shape of veins and arteries are curved in nature and the pumping of blood in these ducts is based on the mechanism of peristalsis.” Comment: The sentence above is faulty. Revise or you may consider, “Curved veins and arteries make up the human cardiovascular system, and the peristalsis process underlies the blood flowing in these ducts.”   Q3. Do you know that Fe3O4 is called Iron(II,III) oxide? Replace in the report. For instance, line 32, it was written as, “iron oxide”   Q4. Line 35, it was written, “The Numerical simulations are performed by using the computational software Mathematica built-in ND numerical procedure.” Comment: Delete the preposition, “by” it is redundant.   Q5. Line 40 – 41, it was written, “dimension less”. Change to dimensionless   Q6. Line 37 – 38, it was written, “The comprehensive graphical observations are presented to see the influence of involved parameters on velocity, temperature, pressure rise, trapping phenomena, and heat transfer coefficient and skin friction.” Comment: Delete. Irrelevant.    Q7. Line 122 – 135 are highly irrelevant. However, It is better to replace with the research questions. Is there any gap between the title, abstract, introduction, discussion of results, and conclusion? This question calls for the introduction of research questions. It is very possible to use the answers to make the report/manuscript to be a user-friendly product. Firstly, the suggestions by Adrian Wallwork in a book titled, “English for Writing Research Papers. Springer New York Dordrecht Heidelberg London, 2011. doi: 10.1007/978-1-4419-7922-3” also encourage the introduction of such question. Secondly, the author should see Fig. 2 to Fig. 9 as typical answers to unknown questions. This is true because the manuscript provides some powerful answers to unknown questions. The author should update the manuscript with appropriate and relevant research questions. This would help readers to link what is known in the literature with the novelty of this study. The research questions may be stated at the end of the introduction. However, a new subsection may be raised to pose the research questions.   Q8. There are three major components of a paragraph under the introduction section. The components are the definition of the term, theoretical review telling us about published aim, and empirical review telling us published results within the scope of the subject matter. To my surprise, the authors were unable to strike a balance. You may check the video links below to see typical examples of theoretical and empirical reviews on the significance of the heater's size, internal heating, and heat sink. https://youtu.be/ugFJnflnsF0 Comment: Revise each paragraph of the introduction section to accurately present the definition, the theoretical, and empirical reviews of the major keywords.   Q9. The conclusion section is not meant for sharing of observation. For instance, line 347, it was written, “The enhancing in nanoparticles concentration, the lower velocity trend is observed.” The conclusion section should be revised. Try to itemize all the conclusive facts. In a short Conclusion, state the most important outcome of the work based on the interpreted findings. Do NOT just summarize. Comment: Report your success in addressing the research questions.   Q10. Update Subsection 4.4 with the fact that: The Nusselt numbers for a fluid flow, according to Ref. A et al. (2022), represent the amount of heat transmission across the layer via pure conduction. For instance, a higher Nusselt number indicates that convection is operating more effectively.

Ref. A et al. (2022):::::: Animasaun, I. L., Shah, N. A., Wakif, A., Mahanthesh, B., Sivaraj, R., & Koriko, O. K. (2022). Ratio of Momentum Diffusivity to Thermal Diffusivity: Introduction, Meta-analysis, and Scrutinization. Chapman and Hall/CRC. New York. ISBN-13: 978-1032108520, ISBN-10: 1032108525, ISBN9781003217374. https://doi.org/10.1201/9781003217374

Author Response

Response to the Reviewer #2

The authors are thankful to the anonymous reviewer for useful comments. These suggestions really enhance the quality of the manuscript. The detail response to the reviewer’s suggestions is as follows.

Reviewer 02

The report presents the outcome of a study on blood conveying Iron(II,III) oxide and Copper nanoparticles through tapered complex wavy curved channel with slip features. The contribution of the report to the body of knowledge is significant. Noteworthy that the objectives of the research study are within the scope of Micromachines. But, the present form of the report needs revision using the comments and the questions/observations listed below  

Q1. The present form of the title is not acceptable. Revise or consider: Exploration of blood conveying Iron(II,III) oxide and Copper nanoparticles through tapered complex wavy curved channel with slip features  

Response: Now title is improved.

Q2. Abstract, line 28, it was written, “In the human cardiovascular system, the shape of veins and arteries are curved in nature and the pumping of blood in these ducts is based on the mechanism of peristalsis.” Comment: The sentence above is faulty. Revise or you may consider, “Curved veins and arteries make up the human cardiovascular system, and the peristalsis process underlies the blood flowing in these ducts.”  

Response: Thank you. The suggested changes have been done.

Q3. Do you know that Fe3O4 is called Iron(II,III) oxide? Replace in the report. For instance, line 32, it was written as, “iron oxide”  

Response: Thank you again. Its corrected now.

Q4. Line 35, it was written, “The Numerical simulations are performed by using the computational software Mathematica built-in ND numerical procedure.” Comment: Delete the preposition, “by” it is redundant.  

Response: Now its done.

Q5. Line 40 – 41, it was written, “dimension less”. Change to dimensionless  

Response: Corrected.

Q6. Line 37 – 38, it was written, “The comprehensive graphical observations are presented to see the influence of involved parameters on velocity, temperature, pressure rise, trapping phenomena, and heat transfer coefficient and skin friction.” Comment: Delete. Irrelevant.   

Response: Now done.

Q7. Line 122 – 135 are highly irrelevant. However, It is better to replace with the research questions. Is there any gap between the title, abstract, introduction, discussion of results, and conclusion? This question calls for the introduction of research questions. It is very possible to use the answers to make the report/manuscript to be a user-friendly product. Firstly, the suggestions by Adrian Wallwork in a book titled, “English for Writing Research Papers. Springer New York Dordrecht Heidelberg London, 2011. doi: 10.1007/978-1-4419-7922-3” also encourage the introduction of such question. Secondly, the author should see Fig. 2 to Fig. 9 as typical answers to unknown questions. This is true because the manuscript provides some powerful answers to unknown questions. The author should update the manuscript with appropriate and relevant research questions. This would help readers to link what is known in the literature with the novelty of this study. The research questions may be stated at the end of the introduction. However, a new subsection may be raised to pose the research questions.  

Response: Now done.

Q8. There are three major components of a paragraph under the introduction section. The components are the definition of the term, theoretical review telling us about published aim, and empirical review telling us published results within the scope of the subject matter. To my surprise, the authors were unable to strike a balance. You may check the video links below to see typical examples of theoretical and empirical reviews on the significance of the heater's size, internal heating, and heat sink. https://youtu.be/ugFJnflnsF0 Comment: Revise each paragraph of the introduction section to accurately present the definition, the theoretical, and empirical reviews of the major keywords.  

Response: Now the intorudction section is modified.

Q9. The conclusion section is not meant for sharing of observation. For instance, line 347, it was written, “The enhancing in nanoparticles concentration, the lower velocity trend is observed.” The conclusion section should be revised. Try to itemize all the conclusive facts. In a short Conclusion, state the most important outcome of the work based on the interpreted findings. Do NOT just summarize. Comment: Report your success in addressing the research questions.  

Response: Now re-phrased and improved.

Q10. Update Subsection 4.4 with the fact that: The Nusselt numbers for a fluid flow, according to Ref. A et al. (2022), represent the amount of heat transmission across the layer via pure conduction. For instance, a higher Nusselt number indicates that convection is operating more effectively.

Ref. A et al. (2022):::::: Animasaun, I. L., Shah, N. A., Wakif, A., Mahanthesh, B., Sivaraj, R., & Koriko, O. K. (2022). Ratio of Momentum Diffusivity to Thermal Diffusivity: Introduction, Meta-analysis, and Scrutinization. Chapman and Hall/CRC. New York. ISBN-13: 978-1032108520, ISBN-10: 1032108525, ISBN9781003217374. https://doi.org/10.1201/9781003217374

Response: This study is included, ref [37].

Round 2

Reviewer 1 Report

Micromachines - 1877929

Comments from Reviewer Round 2

Overall, the revised paper has been improved significantly. Most of the comments raised have been addressed satisfactorily. However, there are still doubts over some of my previous comments as follows.

Comment No. 4, please explain why studying the shear thinning and shear thickening effects associated the human blood is important? You may relate it with specific applications to demonstrate its importance. The points for novel aspects are too lengthy. Consider reducing it.

Comment No. 6, I don’t see any changes made. There is no mention of table 2 and the purpose of presenting table 2.

Comment no. 9, if there is no validation for the theoretical model, the validity of the results is doubtful. As a suggestion, authors may consider developing a theoretical model prior to the actual model and validate it against other published works.

Comment no. 12, I don’t see any changes made to Tables 3 and 4.

Comment no. 15, I still see some errors within the content of the paper like “::” Please check through the content of the paper carefully.

Also, please indicate where the revision has been made (i.e., Page XX, Section YY, Lines ZZ) so it is easier for reviewer/editor to follow.

Author Response

Response to the Reviewer #1

The authors are thankful to the anonymous reviewer for useful comments. These suggestions really enhance the quality of the manuscript. The detail response to the reviewer’s suggestions is as follows.

Overall, the revised paper has been improved significantly. Most of the comments raised have been addressed satisfactorily. However, there are still doubts over some of my previous comments as follows.

 Comment No. 4, please explain why studying the shear thinning and shear thickening effects associated the human blood is important? You may relate it with specific applications to demonstrate its importance. The points for novel aspects are too lengthy. Consider reducing it.

Response: In fact, the human blood characterized the important of shear thinning and shear thickening consequences. The Casson fluid model also attained the same rheological features. Therefore, we have used Casson fluid model for justifying the blood characteristics. Now novelty of work is reduced and summarized in following form:

The motivations for presenting the current flow model are observing the thermal impact of hybrid nanofluid model for slip flow of Casson fluid with applications of peristaltic phenomenon [34-36]. The motivations for considering the Casson fluid model as justified as it reports the shear thinning and shear thickening effects associated the human blood. The novel aspects of current research are:

  • Present a mathematical model for the peristaltic transport of Casson fluid with interaction of hybrid nanofluid containing the ferro nanoparticles and copper nanomaterials in curved channel.
  • The role of slip effects and Hall current is also observed.
  • The highly nonlinear system of obtained model is numerical solved with ND-Solver.
  • The physical thermal impact of hybrid nanoparticles is focused to control the blood flow properties. Current investigation present novel applications for human blood flow, thermal systems, various engineering processes, extrusion systems, human endoscopy, control of heating phenomenon, chemical processes and biomedical applications [37-42].

These explanations are added on page 4.

Comment No. 6, I don’t see any changes made. There is no mention of table 2 and the purpose of presenting table 2.

  Response: Table 2 aims to presents the hybrid nanofluid characteristics in mathematical forms. Now we have assigned proper reference to this table (see page 6).

Table 2: The properties of ferro nanoparticles, copper and human blood [5, 16].

Material

Blood

8933

1063

5200

401

6

385

3594

670

25000

Comment no. 9, if there is no validation for the theoretical model, the validity of the results is doubtful. As a suggestion, authors may consider developing a theoretical model prior to the actual model and validate it against other published works.

 Response: The comparison of results is presented in table 3 and table 4 for both nanofluid and hybrid nanofluid on page 17 and 18.

Table 3: Variation of heat transfer coefficient with  and .

 nanofluid

Hybrid nanofluid

 nanofluid

Hybrid nanofluid

0.0

1.0

1.0

3.0

1.0

2.0

0.0

1.0

2.0

0.0

2.0

4.0

2.5

5.0

Table 4: Change in skin friction with  and .

 nanofluid

Hybrid nanofluid

 nanofluid

Hybrid nanofluid

0.0

1.0

3.0

1.0

2.0

0.0

1.0

2.0

2.5

5.0

Comment no. 12, I don’t see any changes made to Tables 3 and 4.

 Response: Now minor changes are done in these tables. In fact, the aim of table 3 is to present the comparative numerical values of heat transfer for nanofluid and hybrid nanofluid models. Similarly, table 4 is necessary to justify the numerical observations for skin friction coefficient (page 17 and 18).

Table 3: Variation of heat transfer coefficient with  and .

 nanofluid

Hybrid nanofluid

 nanofluid

Hybrid nanofluid

0.0

1.0

1.0

3.0

1.0

2.0

0.0

1.0

2.0

0.0

2.0

4.0

2.5

5.0

Table 4: Change in skin friction with  and .

 nanofluid

Hybrid nanofluid

 nanofluid

Hybrid nanofluid

0.0

1.0

3.0

1.0

2.0

0.0

1.0

2.0

2.5

5.0

Comment no. 15, I still see some errors within the content of the paper like “::” Plase check through the content of the paper carefully.

 Response: Some typo errors are now corrected. 

Also, please indicate where the revision has been made (i.e., Page XX, Section YY, Lines ZZ) so it is easier for reviewer/editor to follow.

Response: Now indicated in revised draft. Thank you.
